# A Research on the Evaluation of China's Food Security under the Perspective of Sustainable Development—Based on an Entropy Weight TOPSIS Model

**Xiaoyun Zhang** [1] , **Yu Wang** [1], **Jie Bao** [1], **Tengda Wei** [2,*] **and Shiwei Xu** [1,*]

[1]   Agricultural Information Institute, Chinese Academy of Agricultural Science, Beijing 100081, China
[2]   College of Economics and Management, China Agricultural University, Beijing 100083, China
*   Correspondence: weitengda@cau.edu.cn (T.W.); xushiwei@caas.cn (S.X.)

**Abstract:** Food safety is an important basis for promoting economic development, ensuring social stability and maintaining national security. Research on the evaluation of food security is the basis by which to accurately grasp the food security situation and to establish national food security policies in a scientific manner. Based on China's agricultural economic data from 2001 to 2020, this research uses an entropy weight TOPSIS model to start from the new connotations and goals of food security in the new development stage, takes quantity security, structural security, ecological security of resources, economic security and policy security as the breakthrough points, builds a food security evaluation system containing 25 indicators, and aims to conduct evaluation and research on the evolution and current situation of China's food security. The results show that China's food security level drops first and then rises, that China attaches increasing importance to the ecological security and policy security of food resources, and that China's food security level is restricted by such factors as resources, modes of production, circulation, storage, transportation, trade and structure. This article puts forward some policies and suggestions in terms of resources, technology and foreign trade to safeguard China's food security.

**Keywords:** food security; sustainable development; indicator system; evaluation; entropy weight TOPSIS model

## 1. Introduction

Food safety is an issue of overall national strategy and an important basis to promote economic development, safeguard social stability and maintain national security, and it has been a concern in all walks of life for a long time [1,2]. China has always attached great importance to food safety, and has introduced a series of policies such as "taking grain as the key link and ensuring an all-round development", "ensuring basic self-sufficiency of grain and absolute security of staple food" and "establishing 'great food security concept'", which fully reflects the degree of importance attached by China to food security in the long term [3]. In recent years, China has had a succession of good harvests in terms of grain production, and its grain circulation capacity has greatly improved, so that the 1.3 billion Chinese people have no need to worry about the problem of food, but at the same time, several new conditions and problems arose, presenting a formidable challenge to food security work. From the perspective of production links: first, the environmental constraints of resources such as arable land and freshwater resources are aggravated, and the constraints of sustainable development of food are increasing; second, production becomes divorced from marketing, and the supply of some varieties can become seriously excessive in stages; third, the cost of grain planting rises, high yield co-exists with low efficiency, farmers' incomes can hardly be increased, and the financial burden on farmers becomes heavy [4,5]. From the perspective of demand: first, with the changes in the childbearing policy and the acceleration of the urbanization process, the food demand increases rapidly,

and this trend will not change; second, with the increase in residents' income level, the grain consumption increase is accelerated, and the contradictions in varieties and quality structures become prominent [6]. Moreover, such factors as globalization, climatic variation, ecological degradation, COVID-19 pandemic, regional conflicts and carbon peaking and carbon neutrality goals also present significant challenges to China's food security [7–9].

Food security is a relative and dynamic concept. The state of food security (or insecurity) can be reflected through a certain number of indicators, to answer the questions such as "what's food security" and "how to evaluate the development level of food security" [10]. The evaluation and research on food security is not only the basis to grasp the situation of food security, but also the basis for a state to formulate food security policies [11,12]. No.1 Central Document for 2022 clearly points out that China should fully implement "equal accountability of Party and government organizations" in terms of food security, strictly carry out the examination of the food security responsibility system, vigorously carry out green actions with high quality and high efficiency, deeply implement high-quality food projects, and fully improve the level of food security. Against this background, the scientific design of the food security evaluation indicator system and the objective measurement of China's food security level are major issues in determining the development direction of China's food security, reducing China's food security risks and building a sustainable agriculture and food system [13,14].

The Food and Agriculture Organization of the United Nations (FAO) has defined food security three times, and these definitions of food security are widely accepted by the international community. The essence of food security is to enable everyone to enjoy sufficient food at any time and to meet the living needs of people. On this basis, many experts, scholars and related agencies in China and abroad adopted multiple methods to conduct quantitative evaluation of food security [15–17]. In 2013, the Food and Agriculture Organization (FAO) released the "2013 Report on the State of Food Insecurity in the World: Multiple Dimensions of Food Security", in which they selected 30 indicators from four dimensions, i.e., food availability, food stability, food access and food utilization, built a food security evaluation indicator system, and conducted the evaluation of the state of food security in 157 countries and regions in the world [18]. After that, there were many foreign researchers who established food security evaluation systems from the aforementioned four dimensions to conduct evaluation of the state of food security in their countries or regions [19–24]. Moreover, there were many foreign researchers who started from the macroscopic angle and adopted the questionnaire method to conduct comprehensive evaluation of the state of food security in countries or regions from the microscopic angle [25–30].

Chinese scholars mostly adopt multi-indicator comprehensive measurement methods to conduct evaluations of food security. The research thought is as follows: building a multi-stage indicator system first, conducting dimensionless standardized processing of indicators, calculating the development levels of indicators one by one, and finally calculating comprehensive indexes on the basis of different weights. For example, Wang Guomin et al. (2013) [31] adopted the Delphi method and the AHP method to establish a food security evaluation system containing nine indicators, and measured China's food security level. Yao Chengsheng et al. (2015) [32] built China's food security evaluation indicator system from four levels, i.e., grain production resources, food availability and stability, food access and food utilization, and conducted quantitative evaluation of the state of food security in China. Mao Xuefeng et al. (2015) [33] evaluated the state of food security in China from the perspectives of food structure, food circulation and trade links. Gao Yanlei et al. (2019) [34] started from major grain producing areas, and adopted the entropy weight TOPSIS model to evaluate the state of food security in China from four dimensions, i.e., availability, access, stability and sustainability. Furthermore, Cui Mingming et al. (2019) [35], Li Xiuxiang et al. (2020) [36], and Qi Yue et al. (2020) [37] applied different methods of evaluating food security through building different indicator systems.

Generally speaking, there are rich research results regarding food security, which can reflect some of the characteristics of food security. However, there are some shortcomings in the research process, such as limited sample selection, single or crossed indicator selection, and strong subjectivity, so that the research can hardly reflect the actual development level of food security in a comprehensive, systematic and accurate manner. As China steps into a new development stage and the environment of food security both at home and abroad changes fundamentally, the connotation of food security is steadily enriched and the goals of ensuring food safety are constantly boosted [1]. The previous research studies on the food security indicator system are more focused on quantity security, but less focused on structural security, ecological security of resources, economic security and policy security, so that they cannot meet the needs of evaluation of food security level under the current development strategy. On the basis of previous research, both at home and abroad and in combination with the new connotation and goals of the food security concept in the new era, this article establishes a food security evaluation system from five levels, i.e., quantity security, structural security, ecological security of resources, economic security and policy security, and adopts the entropy weight TOPSIS model to conduct quantitative evaluation of China's food security. Moreover, it reviews the historical evolution of China's food security on the basis of evaluation results, analyzes the shortcomings and potential risks existing in China's food security, and puts forward corresponding policy suggestions.

Compared with the existing literature, this article may have the following marginal contributions: (1) the food security evaluation indicator system built in the research is more comprehensive and systematic, and when considering quantity security, the system is more focused on such important aspects as structural security, ecological security of resources, economic security and policy security; (2) the entropy weight TOPSIS model used in the research can avoid subjective bias effectively and improve the accuracy of judgment about the actual development level of food security; (3) in the research reviews the development and evolution of food security on the basis of the evaluation results, determines the root cause of food security issues, and provide specific recommendations for China to solve food security issues.

The structure arrangement of the remainder of this article is as follows: Part 2 is concerned with the construction and research design of the food security evaluation indicator system, including data sources, modeling principles and modeling steps; Part 3 presents empirical research, focused on the analysis of model results; Part 4 mainly discusses the evolution of the development level of food security, the existing problems, the shortcomings of the research, and the next research direction; Part 5 sets forth conclusions and policy suggestions.

## 2. Methods and Data

### 2.1. Construction of Evaluation Indicator System

2.1.1. Design Principles for Indicator System

Food security issues involve multiple factors. All these factors are interrelated and interact with each other, and constitute an organic whole. Due to the systematic nature and complexity of food security influencing factors, the food security evaluation indicator system should be a well-bedded and well-organized complex. Thus, the setting of the food security evaluation indicators should conform to the following principles:

(1) Systematic Principle. The food security evaluation indicator system is a system concept, consisting of the subsystems at different layers, such as target layer, criterion layer and indicator layer. All system layers are interrelated, depend on each other, influence each other and restrict each other. When building the food security evaluation indicator system, one is required to adopt systematic thinking, i.e., regarding the evaluation indicator system as an effective, inclusive and open system that is characterized by the natures of dynamics and complexity. Furthermore, it is a requirement to apply system-related theories to carrying out the overall layout of the evaluation indicator system and realize the

optimization and upgrading of the organic whole through interaction and transformation of the factors inside and outside the system [38,39].

(2) Scientific Principle. The design of the food security evaluation indicator system must stick closely to the stage goals, contemporary connotation and features of food security and carry out evaluation from the aspects such as economic benefits, social benefits and ecological benefits. The evaluation indicators should be able to reflect the actual development level of food security in the current stage as well as the future development direction and development potential of food security. It is a requirement to define the category and weight of each indicator in the evaluation system in a scientific and reasonable manner, and select and use scientific calculation methods and models to carry out quantification and evaluation [40].

(3) Guiding Principle. The design of the food security evaluation indicator system should fully reflect the emphasis and key points of the construction of food security in countries or regions, and it is the breakthrough and point of strength to improve the food security level at present and in the short run. Through the construction and use of the food security evaluation indicator system, directional guidance should be provided for countries or regions to improve their food security levels, enabling academic circles and related government decision-making departments to clearly understand the focus and direction of future research [30,38].

(4) Operability Principle. The design of the food security evaluation indicator system should be in line with the development goals set forth in relevant policy documents such as No.1 Central Document formulated by the Party and the state as well as Outline of the 14th Five-Year Plan (2021–2025) for National Economic and Social Development and Vision 2035 of the People's Republic of China, to ensure that the evaluation indicator system highlights major points and has a clear logic and a reasonable framework. Each designed indicator should be simple, clear and measurable, and all involved data should be of strong availability, to facilitate future calculation and evaluation [37,38,41].

(5) Harmonization Principle. The design of the food security evaluation indicator system shall harmonize the comprehensiveness and representativeness of evaluation indicators. The evaluation indicator system should reflect the development state of food security as fully and comprehensively as possible, and cover the essence of the connotation of food security in the current stage. However, the food security evaluation indicator system is unlikely to cover every aspect, and can only involve important areas and representative areas; thus, it is a requirement to find the point of equilibrium between full coverage and representativeness. Moreover, the design of the evaluation indicator system requires coordinating the relationship among the past, the present and the future, and it does not only need to meet the current needs and consider the future condition, but also need to adopt the existing evaluation indicator system [38,42].

### 2.1.2. Construction of Indicator System

In the construction process of the food security (A) evaluation system, the research regards "sustainable development" as the main line, sticks closely to the contemporary connotation of food security, and follows the systematic principle, the scientific principle, the guiding principle, the operability principle and the harmonization principle. On the basis of related research results both at home and abroad, corresponding indicators can be selected to build a food security evaluation system from 5 aspects, i.e., quantity security (A1), structural security (A2), ecological security of resources (A3), economic security (A4) and policy security (A5). The system sets 25 indicators in total, including volatility of grain yield, grain sowing area and grain yield per unit area (Table 1). Evaluation indicators consist of positive indicators and negative indicators. Positive indicators are positively correlated with the level of food security.

**Table 1.** Food Security Evaluation Indicator System.

| 1st Grade Indicator | 2nd Grade Indicator | Unit | Properties |
|---|---|---|---|
| Quantity Security (A1) | Volatility of Grain Yield (A11) | % | Negative |
| | Grain Sown Area (A12) | 1000 ha. | Positive |
| | Grain Yield per Unit Area (A13) | kg/ha. | Positive |
| | Per Capita Grain Possession (A14) | kg/person | Positive |
| Structural Security (A2) | Degree of Dependence on Grain Foreign Trade (A21) | % | Positive |
| | Proportion of Feed Grain Sowing Area in Grain Sowing Area (A22) | % | Positive |
| | Proportion of Soybean Imports in Grain Imports (A23) | % | Negative |
| | Stock-to-Use Ratio (A24) | % | Positive |
| Ecological Security of Resources (A3) | Pesticide Consumption per Unit Sown Area (A31) | kg/ha. | Negative |
| | Consumption of Chemical Fertilizers per Unit Sown Area (A32) | kg/ha. | Negative |
| | Proportion of Effective Irrigation Area (A33) | % | Positive |
| | Multiple Cropping Index (A34) | % | Positive |
| | Per Capita Water Resources (A35) | $m^3$/person | Positive |
| | Arable Land Per Capita (A36) | $m^2$/person | Positive |
| | Proportion of Disaster-affected Area (A37) | % | Negative |
| Economic Security (A4) | Food Price Volatility (A41) | % | Negative |
| | Net Profit of Grain Planting (A42) | RMB/ha. | Positive |
| | Engel's Coefficient of Rural Residents (A43) | % | Negative |
| | Agricultural Labor Productivity (A44) | 10,000 RMB/Person | Positive |
| | Agricultural Land Productivity (A45) | 10,000 RMB/ha. | Positive |
| Policy Security (A5) | Transportation Route Intensity (A51) | km/km$^2$ | Positive |
| | Agricultural Mechanization Level (A52) | kW/ha. | Positive |
| | Contribution Rate of Agricultural Scientific and Technological Progress (A53) | % | Positive |
| | Educational Level of Agricultural Labor Force (A54) | % | Positive |
| | Financial Expenditure for Grain Production (A55) | 100 million RMB | Positive |

### 2.1.3. Indicator Calculation and Description

(1) Quantity Security. Quantity security embodies the grain amount that is maintained to ensure food supply capacity and meet people's grain demand. The first indicator designed is volatility of grain yield. This indicator is an important indicator that reflects the stability of grain supply. The computational formula is: $R_t = \left(Y_t - Y_t'\right)/Y_t'$, among which, $Y_t$ represents the grain yield of the year t and $Y_t'$ represents the trend yield, which is expressed as the moving average of five years in this research [32,35]. The second indicator is grain sown area. The grain sown area is the basis of food quantity security. The higher the value, the higher the guaranteed level of food quantity security [36]. The third indicator is grain yield per unit area. This indicator r mainly reflects the degree of development of food technology and efficiency. The higher the value, the higher the guaranteed level of food quantity security [43,44]. The fourth indicator is per capita grain possession. This indicator starts from the microcosmic angle of view and can reflect both the stability of total grain output and the changes in grain supply capacity with the increase in population [45,46].

(2) Structural Security. The food structure in the research contains supply structure, plantation structure and import structure. The first indicator designed is the degree of dependence on grain foreign trade, which is expressed as the ratio of grain import volume to grain yield (in the research, food/grain mainly refers to cereals). This indicator mainly considers the impact of the international grain market on national food security and reflects the degree of structural security of the food supply [47]. The second indicator is the proportion of feed grain sowing area in the grain sowing area. In the research, feed grain mainly refers to corn and soybeans [36]. As people's demand for animal products such as meat, eggs and milk increases, the feed grain consumption will rise accordingly. This indicator is designed to evaluate the state of China's grain planting structure and further reflect the degree of security of the grain planting structure. The third indicator is the proportion of soybean imports in grain imports. The design of this indicator is mainly

based on China's large soybean imports and high degree of dependence upon foreign trade, and this indicator is used to evaluate the degree of structural security of grain import. In this indicator, grain mainly refers to rice, wheat, corn and soybeans. The fourth indicator is stock-to-use ratio [48]. This indicator mainly embodies the food supply capacity of the state in responding to major natural disasters, wars and other serious sudden events. The calculation formula is $\beta = (S_t / C_{t+1}) * 100\%$, among which, $\beta$ represents stock-to-use ratio, $S_t$ represents the carry-over stock of the year t, and $C_{t+1}$ represents the grain consumption of the year $t + 1$.

(3) Ecological Security of Resources. The indicator system mainly reflects the connotation and characteristics of the sustainability of food security and has received extensive attention. The two indicators designed refer to pesticide consumption per unit sown area and consumption of chemical fertilizers per unit sown area, which are, respectively, expressed as the ratio of pesticide consumption and the ratio of chemical fertilizer consumption to sown area of farm crops. The excessive use of chemical fertilizers and pesticides would lead to serious agricultural non-point source pollution, has a big impact on arable land and water resources, and further restricts the sustainability of grain production [49,50]. The two indicators, i.e., proportion of effective irrigation area [51] and multiple cropping index [52], are designed to evaluate the sustainability of grain production from the perspective of resource utilization efficiency. The proportion of effective irrigation area is expressed as the ratio of effective irrigation area to sown area of farm crops. This multiple cropping index reflects the degree of reutilization of arable resources within one year, and it is expressed as the ratio of sown area of farm crops to arable area. The two indicators, i.e., per capita water resources and arable land per capita, are designed to evaluate the supply status of the main resources required by grain production from the microscopic perspective, and they are, respectively, expressed as the ratio of water resource quantity to population and the ratio of arable area to population [53]. Plant diseases, insect pests and natural disasters are the results of the interaction among the species in the farmland ecosystem and the interaction between crops and climatic conditions, and constitute an important indicator used to evaluate the ecological environment security of grain. The research adopts the proportion of disaster-affected area to reflect the indicator [54]. The calculation formula is $R_d = S_d / S * 100\%$, among which, $R_d$ represents the proportion of disaster-affected area, $S_d$ represents the disaster-affected area, and $S$ represents the total sown area.

(4) Economic Security. This indicator system focuses primarily on consumption, effectiveness and efficiency. The first indicator designed is food price volatility. Food price fluctuation is the result of interaction between various factors, and can reflect the overall risk faced by food security in a comprehensive manner [31]. In order to eliminate the impact of inflation on food price volatility, the calculation formula of the indicator used in the research is $\varnothing = (GPI / CPI - 1) * 100\%$, among which, $\varnothing$ represents food price volatility, $GPI$ represents grain price index, and $CPI$ represents consumer price index. The second indicator is net profit of grain planting [55]. The profit of grain planting directly affects the direct income of farmers and influences the enthusiasm of farmers for grain planting. The higher the profit of grain planting and the enthusiasm of farmers for grain planting, the higher the guaranteed extent of food security. The research adopts the net profit of grain production per hectare to evaluate the profit of grain planting of farmers, which mainly refers to the net profit of staple food grain planting. Considering the urban–rural income gap and the availability of statistical data, Engel's coefficient of rural residents is regarded as one of the evaluation indicators [47]. Engel's coefficient reflects the proportion of residents' food expenses in their living expenses. It is negatively correlated to the food security level and it is an indicator designed to measure the fairness in food security. Agricultural labor productivity and agricultural land productivity are designed to evaluate the effectiveness and efficiency of grain output and then evaluate the economic sustainability of food security [47,55]. Agricultural labor productivity is expressed as the ratio of gross agricultural output value to the number of employed persons of the primary

industry; agricultural land productivity is expressed as the ratio of gross agricultural output value to the sown area of farm crops.

(5) Policy Security. This indicator system focuses primarily on all kinds of policies and safeguards supporting food security, e.g., financial expenditure, infrastructures, technological development and talent cultivation. The first indicator is transportation route intensity [32]. Road traffic is closely related to grain production, allocation and transportation, and it is the basic condition for equilibrium in the supply of grain. This indicator is mainly used to measure the availability of grain and is expressed as the length of transportation route per unit area. The second indicator is agricultural mechanization level [56]. This indicator reflects the degree of mechanization of grain production, and it is used to measure the production sustainability and efficiency of food security and is expressed as the ratio of total power of agricultural machinery to sown area of farm crops. The third indicator is contribution rate of agricultural scientific and technological progress [47]. The sustainable development of food security must be supported by modern agricultural science and technology as well as advanced materials and equipment. On the condition that technical conditions remain unchanged, scientific and technological progress is an important driver to promote the improvement of grain input–output level, so that the evaluation indicator is set. The fourth indicator is the educational level of the agricultural labor force [57]. The agricultural labor force is the main body of grain production. The quality of the agricultural labor force directly determines the level of production of grain, and thus influences the development level of national food security. In the research, this indicator is expressed as the number of technical secondary school or college graduates or above in every 100 rural workers. The fourth indicator is financial expenditure for grain production [35]. The support and regulation of the government for food security is reflected in multiple links such as production, allocation and storage, and the measures adopted are varied, including grants, subsidies and technical support, and can hardly be measured in a comprehensive manner. Considering that the strength of support and regulation is generally reflected in financial input, the research adopts financial expenditure for grain production as an evaluation indicator. The calculation thought of this indicator is to reduce the state financial expenditure on agriculture, forestry and water affairs according to the proportion of grain sown area to total sown area. The calculation formula is $B_f = F \times S_f / S$, among which, $B_f$ represents financial expenditure for grain production, $F$ represents financial expenditure for agriculture, forestry and water, $S_f$ represents grain sown area, and $S$ represents total sown area of farm crops.

### 2.2. Data Sources

Subject to the data release condition and the availability of indicator data, the research adopts national grain production, consumption, prices, resource environment and other factors for the period from 2001 to 2020 as its objects, and aims to conduct evaluation and research of the development level of China's food security on this basis. The data used by various food security evaluation indicators of the research are, respectively, sourced from the statistical yearbooks, such as *China Statistical Yearbook*, *China Rural Statistical Yearbook*, *National Agricultural Products Cost–benefit Data Collection*, *China Grain Yearbook*, *China Yearbook of Agricultural Price Survey* and *China Statistical Yearbook on Environment*, as well as the statistical information published by the website of Ministry of Agriculture and Rural Affairs, the website of National Bureau of Statistics and the website of General Administration of Customs. Moreover, some indicator values are calculated according to relevant data of the BRIC Agricultural DataBase (Table 2).

### 2.3. Entropy Weight TOPSIS Model

#### 2.3.1. Model Introduction

The TOPSIS method is a multi-attribute decision-making method with finite alternatives, which is an extremely important method in multi-objective decision analysis, also called the "Technique for Order Preference by Similarity to Ideal Solution". This method

was first put forward by Hwang CL and Yoon K in 1981 [58]. The essence of the entropy weight TOPSIS model is the improvement of the traditional TOPSIS method. This method is mainly utilized by determining the weights of all evaluation indicators on the basis of the entropy weight method, effectively eliminating the deviations in evaluation indicator weights caused by subjective factors, and further making use of the technique for similarity to the ideal solution to determine the sort order of evaluation objects [38].

**Table 2.** Data Sources.

| 1st Grade Indicator | 2nd Grade Indicator | Data Sources |
| --- | --- | --- |
| A1 | A11 | Calculated according to the relevant data of "*China Statistical Yearbook*" |
| | A12 | *China Statistical Yearbook* |
| | A13 | National Bureau of Statistics |
| | A14 | National Bureau of Statistics |
| A2 | A21 | Calculated according to relevant data of BRIC Agricultural DataBase |
| | A22 | Calculated according to the relevant data of "*China Statistical Yearbook*" |
| | A23 | Calculated according to the relevant data on the website of the General Administration of Customs |
| | A24 | Calculated according to relevant data of BRIC Agricultural DataBase |
| A3 | A31 | Calculated according to the relevant data of " *China Rural Statistical Yearbook* " |
| | A32 | Calculated according to the relevant data of " *China Rural Statistical Yearbook* " |
| | A33 | Calculated according to the relevant data of the National Bureau of Statistics |
| | A34 | Calculated according to the relevant data of the National Bureau of Statistics |
| | A35 | *China Yearbook of Agricultural Price Survey* and *China Statistical Yearbook on Environment* |
| | A36 | National Bureau of Statistics |
| | A37 | Calculated according to the relevant data of the National Bureau of Statistics |
| A4 | A41 | *China Grain Yearbook* and *China Yearbook of Agricultural Price Survey* |
| | A42 | *National Agricultural Products Cost–benefit Data Collection* |
| | A43 | National Bureau of Statistics |
| | A44 | Calculated according to the relevant data of the National Bureau of Statistics |
| | A45 | Calculated according to the relevant data of the National Bureau of Statistics |
| A5 | A51 | Calculated according to the relevant data of "*China Statistical Yearbook*" |
| | A52 | Calculated according to the relevant data of the National Bureau of Statistics |
| | A53 | The website of Ministry of Agriculture and Rural Affairs |
| | A54 | *China Rural Statistical Yearbook* |
| | A55 | Calculatd according to the relevant data of the National Bureau of Statistics |

The basic thought behind this method is as follows: first, determine the ideal solution (the negative ideal solution), that is to say, each attribute value has reached the optimal/worst value in the alternative; then, make a judgment by measuring the relative distance between each evaluation object and the optimal/worst solution, and if the evaluation object is closest to the optimal solution and the farthest from the worst solution, the solution is optimal; otherwise, it is non-optimal [59]. The entropy weight TOPSIS model can make full use of original data and reflect the gaps between different alternatives. This model has no special requirements on sample size and is not disturbed by reference sequence selection, and it has such advantages as intuitive geometric significance, reduced loss of information and flexible operation [34,60].

2.3.2. Modeling Process

Suppose there are *m* evaluation objects and *n* evaluation indicators and $x_{ij}$ is the original data of the *j*th indicator in the *i*th evaluation object, the original evaluation indicator matrix X:

$$X = \begin{vmatrix} x_{11} & x_{12} & \cdots & x_{1n} \\ x_{21} & x_{22} & \cdots & x_{2n} \\ \vdots & \vdots & \ddots & \vdots \\ x_{m1} & x_{m2} & \cdots & x_{mn} \end{vmatrix} \tag{1}$$

(1) Standardized Processing of Data. As each indicator has a different dimension, it is a requirement to carry out standardized processing of the data of each indicator first. With regard to positive indicators, Equation (2) shall apply; with regard to negative indicators, Equation (3) shall apply.

$$y_{ij} = \frac{x_{ij} - minx_{ij}}{maxx_{ij} - minx_{ij}} \tag{2}$$

$$y_{ij} = \frac{maxx_{ij} - x_{ij}}{maxx_{ij} - minx_{ij}} \tag{3}$$

In the formulas, $y_{ij}$ is the standardized value of the $j$th indicator in the $i$th evaluation object, and the standardized matrix $Y$ is worked out after processing:

$$Y = \begin{vmatrix} y_{11} & y_{12} & \cdots & y_{1n} \\ y_{21} & y_{22} & \cdots & y_{2n} \\ \vdots & \vdots & \ddots & \vdots \\ y_{m1} & y_{m2} & \cdots & y_{mn} \end{vmatrix} \tag{4}$$

(2) Calculate the characteristic proportion (contribution degree) of the $i$th> evaluation object ($r_{ij}$) under the $j$th indicator according to Equation (5).

$$r_{ij} = y_{ij} \Big/ \sum_{i=1}^{m} y_{ij} \tag{5}$$

The characteristic proportion matrix R is obtained through the calculation above:

$$R = \begin{vmatrix} r_{11} & r_{12} & \cdots & r_{1n} \\ r_{21} & r_{22} & \cdots & r_{2n} \\ \vdots & \vdots & \ddots & \vdots \\ r_{m1} & r_{m2} & \cdots & r_{mn} \end{vmatrix} \tag{6}$$

(3) Calculate the information entropy of each indicator ($e_j$) according to Equation (7).

$$e_j = -K \sum_{i=1}^{m} r_{ij} ln r_{ij} , \ K = 1/lnm \tag{7}$$

(4) Calculate the weight of each indicator ($w_j$) according to Equation (8).

$$w_j = (1 - e_j) \Big/ \sum_{j=1}^{n} (1 - e_j) \tag{8}$$

(5) Adopt vector normalization method to work out normalization matrix $G$.

$$G = \begin{vmatrix} g_{11} & g_{12} & \cdots & g_{1n} \\ g_{21} & g_{22} & \cdots & g_{2n} \\ \vdots & \vdots & \ddots & \vdots \\ g_{m1} & g_{m2} & \cdots & g_{mn} \end{vmatrix} \tag{9}$$

$$g_{ij} = y_{ij} \Big/ \sqrt{\sum_{i=1}^{m} y_{ij}^2} \tag{10}$$

(6) Build weighted and normalized decision-making matrix $Z$.

$$z_{ij} = w_j * g_{ij} \tag{11}$$

$$Z = \begin{vmatrix} Z_{11} & Z_{12} & \cdots & Z_{1n} \\ Z_{21} & Z_{22} & \cdots & Z_{2n} \\ \vdots & \vdots & \ddots & \vdots \\ z_{m1} & z_{m2} & \cdots & z_{mn} \end{vmatrix} = \begin{vmatrix} w_1 g_{11} & w_2 g_{12} & \cdots & w_n g_{1n} \\ w_1 g_{21} & w_2 g_{22} & \cdots & w_n g_{2n} \\ \vdots & \vdots & \ddots & \vdots \\ w_1 g_{m1} & w_2 g_{m2} & \cdots & w_n g_{mn} \end{vmatrix} \tag{12}$$

(7) Determine the positive ideal solution and negative ideal solution of each indicator. Suppose $z_j^+$ and $z_j^-$ are, respectively maximum and minimum values of the $j$th indicator in the matrix Z:

The positive ideal solution is:

$$Z_j^+ = \left[ z_1^+, z_2^+, \cdots, z_n^+ \right] \ (j = 1, 2, \cdots, n) \tag{13}$$

The negative ideal solution is:

$$Z_j^- = \left[ z_1^-, z_2^-, \cdots, z_n^- \right] \ (j = 1, 2, \cdots, n) \tag{14}$$

(8) Calculate the Euclidean distance from each evaluation object to positive ideal solution and negative ideal solution (degree of separation).

$$d_i^+ = \sqrt{\sum_{j=1}^{n} \left( z_{ij} - z_j^+ \right)^2} \ (1 \leq i \leq m, \ 1 \leq j \leq n) \tag{15}$$

$$d_i^- = \sqrt{\sum_{j=1}^{n} \left( z_{ij} - z_j^- \right)^2} \ (1 \leq i \leq m, \ 1 \leq j \leq n) \tag{16}$$

(9) Calculate the degree of similarity ($S_i$) between each evaluation object and positive ideal solution.

$S_i$ represents the closeness of the food security level of the $i$th evaluation object to the optimal level, generally called "degree of similarity". The value range is (0, 1). The greater $S_i$, the higher the food security level [61,62]. The calculation formula is as below:

$$S_i = \frac{d_i^-}{d_i^+ + d_i^-} \ (1 \leq i \leq m) \tag{17}$$

## 3. Results

### 3.1. Determination of Weights of Indicators

The weight of each evaluation indicator is worked out according to Equations (1)–(8). See Table 2 for details. From Table 3, it can be seen that the weights of quantity security, structural security, ecological security of resources, economic security and policy security among food security indexes are, respectively, 14.62%, 16.71%, 28.52%, 19.03% and 21.12%. Among them, the weights of ecological security and policy security of food resources are the greatest, which indicates that China attaches great importance to the environmental security and policy security of good resources and also objectively reflects the state of China's food security for quite a long time in the past. Figure 1 refers to the histogram of weights of indicators. From Figure 1, it can be seen that Volatility of Grain Yield (A11), Degree of Dependence on Grain Foreign Trade (A21), Pesticide Consumption per Unit Sown Area (A31), Consumption of Chemical Fertilizers per Unit Sown Area (A32), Agricultural Labor Productivity (A44) and Educational Level of Agricultural Labor Force (A54) are important factors that influence the development level of China's food security.

**Table 3.** Information entropies and weights of all evaluation indicators of food security.

| 1st Grade Indicator | Weight of 1st Grade Indicator | 2nd Grade Indicator | Information Entropy | Weight of 2nd Grade Indicator |
|---|---|---|---|---|
| A1 | 14.62% | A11 | 0.9622 | 5.42% |
| | | A12 | 0.9556 | 2.80% |
| | | A13 | 0.9580 | 3.29% |
| | | A14 | 0.9309 | 3.11% |
| A2 | 16.71% | A21 | 0.9554 | 5.12% |
| | | A22 | 0.9342 | 3.30% |
| | | A23 | 0.9481 | 3.42% |
| | | A24 | 0.9451 | 4.87% |
| A3 | 28.52% | A31 | 0.9709 | 5.88% |
| | | A32 | 0.9396 | 5.53% |
| | | A33 | 0.9621 | 3.84% |
| | | A34 | 0.9250 | 4.07% |
| | | A35 | 0.9380 | 2.16% |
| | | A36 | 0.9395 | 4.47% |
| | | A37 | 0.9618 | 2.58% |
| A4 | 19.03% | A41 | 0.9581 | 1.81% |
| | | A42 | 0.9168 | 2.80% |
| | | A43 | 0.9388 | 4.27% |
| | | A44 | 0.9269 | 5.56% |
| | | A45 | 0.9538 | 4.59% |
| A5 | 21.12% | A51 | 0.9206 | 4.48% |
| | | A52 | 0.9254 | 2.83% |
| | | A53 | 0.9652 | 3.11% |
| | | A54 | 0.9756 | 6.16% |
| | | A55 | 0.9423 | 4.54% |

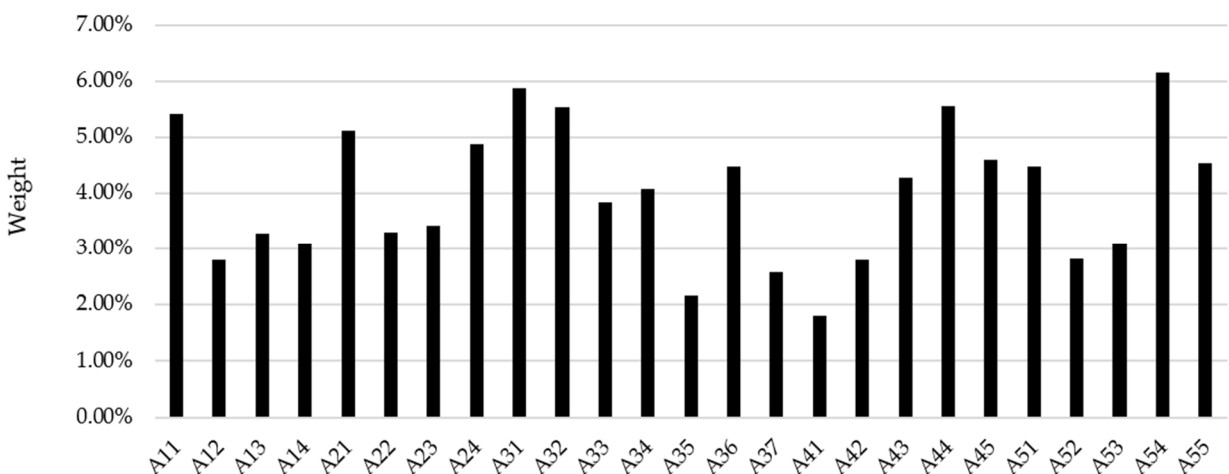

**Figure 1.** Histogram of Weights of Food Security Indicators.

*3.2. China's Food Security Index*

China's food security index and indexes of five second grade indicators for 2001–2020 are calculated on the basis of the abovementioned food security evaluation indicator system and the weights of all indicators (Equations (9)–(17)). The results are shown in Figure 2. Seen from general food security index according to Equation (17), from 2001 to 2007, the state of China's food security had been declining, and after 2007, the state of China's food security began to improve gradually. In terms of the specific ranking of the overall food security index by year, the lowest index level for food security was 0.23 in 2007, after which it began to increase, reaching a maximum value of 0.74 in 2020, as shown in Table 4.

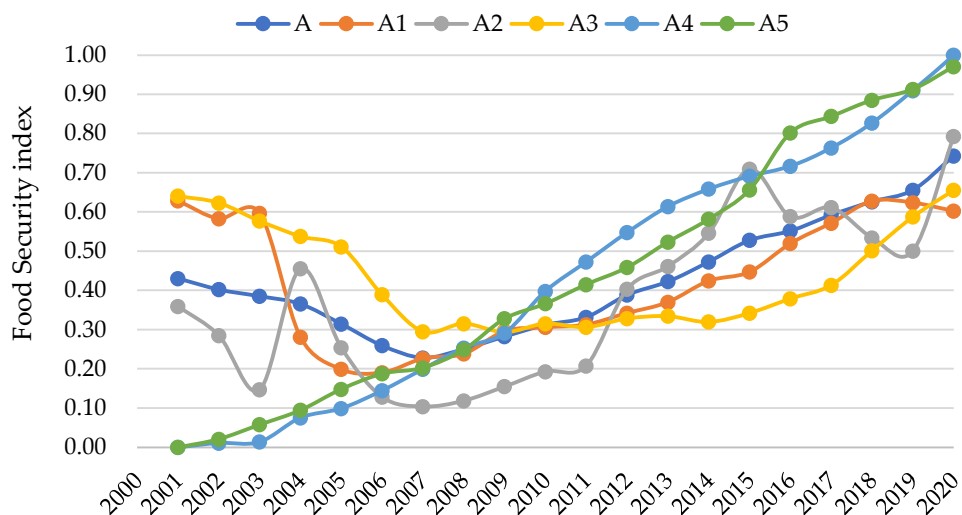

**Figure 2.** Changes in China's Food Security Index from 2001 to 2020.

**Table 4.** Overall food security index and its ranking, 2001–2020.

| Year | Similarity | Rank | Year | Similarity | Rank |
|------|-----------|------|------|-----------|------|
| 2001 | 0.43 | 8 | 2011 | 0.33 | 14 |
| 2002 | 0.40 | 10 | 2012 | 0.39 | 11 |
| 2003 | 0.39 | 12 | 2013 | 0.42 | 9 |
| 2004 | 0.37 | 13 | 2014 | 0.47 | 7 |
| 2005 | 0.31 | 15 | 2015 | 0.53 | 6 |
| 2006 | 0.26 | 18 | 2016 | 0.55 | 5 |
| 2007 | 0.23 | 20 | 2017 | 0.59 | 4 |
| 2008 | 0.25 | 19 | 2018 | 0.63 | 3 |
| 2009 | 0.28 | 17 | 2019 | 0.66 | 2 |
| 2010 | 0.31 | 16 | 2020 | 0.74 | 1 |

As seen from the quantity security index, China's food quantity security showed a trend of growth in fluctuation, i.e., declining firstly and then rising; from 2001 to 2006, the state of China's food security was on the decline, and later, it began to rise gradually. In recent years, the food quantity security index was lower than the general food security index, but the variation trend of the food quantity security index was close to that of the general food security index. This provides support for the general food security index. Seen from the food structure security index, China's food structure security showed a trend of growth in fluctuation on the whole, i.e., declining firstly and then rising, and in some years, the fluctuation range is relatively large, which indicates that there are big risks in China's food structure security.

Seen from the ecological security index of food resources, the variation trend of this index is basically the same as that of the general food security index. From 2007 to 2014, the index rose slightly, and in recent years, it rose rapidly. This is identical to the national food security policy that is focused on greenness, high quality and high efficiency. Seen from economic security index and policy security index, from 2001, both of the indexes were on the rise, and their variation trends of were basically identical. This indicates that the income, living standards and production efficiency of China's rural residents are constantly rising, and reflects the increasing emphasis in China on the technology, policy and fund inputs in terms of agriculture, rural areas and farmers.

*3.3. Changes in China's Food Security Trend*

By developing a radar chart of food quantity security index, structural security index, ecological security index of resources, economic security index and policy security index

(Figure 3), we can analyze the development process of China's food security from 2001 to 2020, which can be generally divided into three stages.

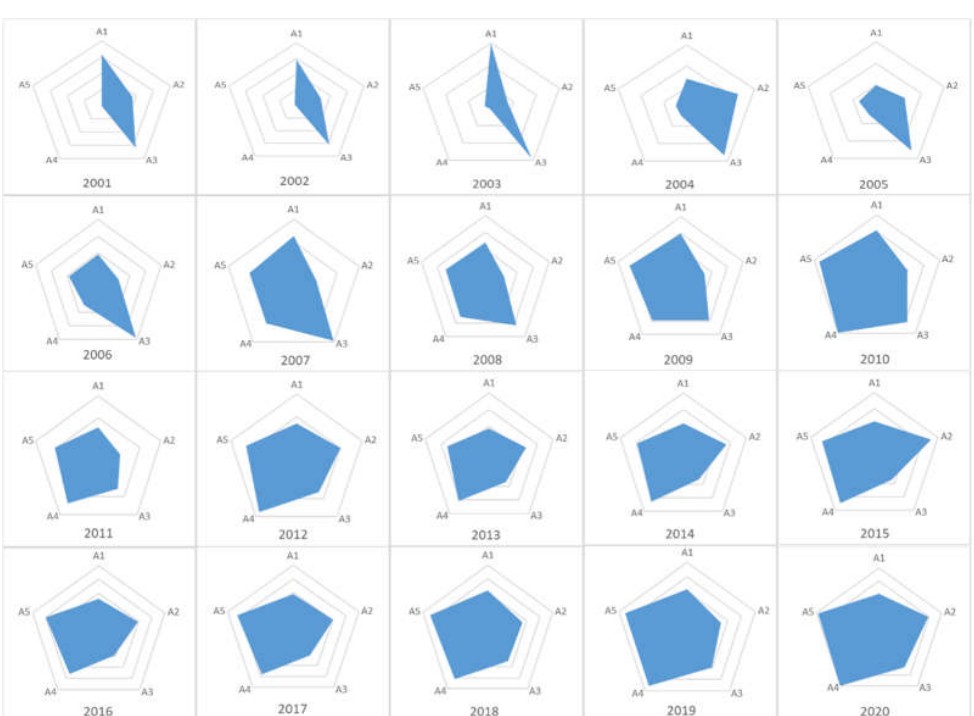

**Figure 3.** Chart of Changes in the Connotation of China's Food Security from 2001 to 2020.

From 2001 to 2006 was a rapid development stage. Food quantity security and ecological security of food resources were at a relatively high level, but economic security and policy security were at a relatively low level. In this stage, China introduced a lot of polies supporting agriculture and benefiting farmers, collection and storage policies and technology policies, and improved the input levels and output levels of factors continuously. The use level of pesticides and chemical fertilizers per unit of sown area was improved continuously, but the overall level was relatively low, the fluctuation range of grain yield was large, and the grain self-sufficiency rate was constantly declining; in the same way, the labor productivity and the agricultural land productivity were rising constantly, but their levels were not high, and so their abilities to support structural security and economic security of food resources were poor.

From 2007 to 2013 was an uneven development stage. Economic security and policy security of food resources were relatively prominent, and structural security of food and ecological security of resources were poor. In this stage, the consumption of pesticides and chemical fertilizers per unit sown area hit a record high, and high inputs in production factors ensured the quantity security of food, but due to the resulting non-point source pollution, the ecological environment security level of food resources continued to decline, grain production seemed to only focus on quantity but not focus on tendency of resource ecology, and the issues relating to agriculture, rural areas and farmer became increasingly prominent. Furthermore, food imports kept growing, but the dependence of soybeans on foreign trade increased year by year, bringing serious hidden troubles to the structural security of food.

From 2014 to 2020 was an even development stage. Food security developed in a balanced manner, and economic security, policy security and ecological security of resources showed a stable and positive trend, but structure security showed a fluctuating downward trend. In this stage, the consumption of pesticides and chemical fertilizers per unit of sown area showed a downward trend, and the ecological security of food resources showed a positive trend. The food structure security situation showed a fluctuating downward

trend and should cause great concern. The main reason might be the increasing demand of residents for animal products, which leads to a sharp increase in the demand for feed grain, but the supply quantity of main feed grains such as corns and soybeans in China is insufficient, and the dependence on importing such feed grains goes up year by year. Moreover, the benefits of grain planting of farmers are low, the land transfer is complex and difficult, and the increase in food inventory aggravates the financial burden, which also brings certain risks to the level of food security.

Generally speaking, all indicators of national food security show a balanced development trend; with the adjustment of national food security strategy and the deepening of food policy reform, the economic security of food will inevitably be improved.

## 4. Discussion

### 4.1. Lack of Arable Land and Fresh Water Resources, Unsustainable Mode of Production

Lack of arable land and freshwater resources is a serious constraint for China's food security [63]. With the acceleration of urbanization and industrialization, China's arable area shows a downward trend. In 2020, China's per capita arable area was only 0.09 hectares per person, 0.17 hectares per person less than the world's per capita arable area (0.26 hectares per person). Meanwhile, the sustained sharp increase in the prices of means of production such as chemical fertilizers leads to the constant rise in the cost of grain production, which is considerably higher than the increase in food prices, and the benefits of grain planting decreased year by year. In 2016, the net profit of grain planting of farmers was −1204.2 RMB/hectare. This was the first time that China suffered a loss in grain planting of farmers. In the three subsequent years, the grain planting of farmers had been at a loss, the enthusiasm of famers for grain planting declined, the phenomenon of non-agricultural circulation of farmland became prominent, and the phenomenon of abandonment of arable land was serious. "Who is going to plant grain in the future" will become a major hidden trouble for China's food security [64]. The freshwater resources are the lifeblood of agricultural production. In 2020, the per capita freshwater quantity in China was 2239.8 m$^3$, only 1/4 of the world average. In terms of the utilization efficiency of water resources, to produce the same amount of food, China needs to use an amount of water equivalent to twice the water consumption in the US. Lack of freshwater resources and inefficient utilization of water resources cannot guarantee stable food supply [35].

The ecological resource environment will directly impact the sustainable supply of food and the sustainable development of agriculture in China. The mode of agricultural production in China involving high investments guarantees the quantity security of food to some extent, but poses a threat to the ecological security and green high quality development of food resources [65]. China is the world's largest country of production and consumption of chemical fertilizers and pesticides. Reasonable use of chemical fertilizers can increase the current grain yield per unit area, but heavy or excessive use of chemical fertilizers would make the ecological environment of the soil and water even worse and make the land productivity decline, accordingly posing a threat to the ecological security of food resources. Excessive use of pesticides would lead to the increase in the level of pesticide residues in crops and the accumulation of pesticide residues in the food chain, thereby reducing the food quality. Since 2015, China has begun to carry out zero growth action in the consumption of chemical fertilizers and pesticides. After the implementation in recent years, the consumption of chemical fertilizers and pesticides decreased significantly, and the utilization rate of chemical fertilizers and pesticides is greatly improved, but the consumption of chemical fertilizers and pesticides per unit area in China is still 3.7 times the world average, which is a significant setback to the sustainable development of China's food resources [1].

### 4.2. High Food Inventory Cost and Excessive Financial Burden

China's high food inventory and big policy-related grain reserves aggravate the financial burden. Take rough rice as an example, if stored for 1 year, the central financial

subsidy is RMB 260 per ton, and for the second year, the central government needs to pay RMB 210 as the storage fee and the interest subsidy. Where a ton of rough rice is stored for 2 years on average, on the basis of the current inventory of rough rice which is more than 100 million tons, the financial expenditure nationwide would exceed RMB 47 billion and cause a huge financial burden. Moreover, under the current grain collection and storage system, the minimum grain purchase price plus the huge grain storage fee makes the grain cost higher [66]. Due to the restriction of the high grain cost, China's grain loses its pricing advantages on the market and cannot effectively use the international market to reach a balance in the throughput, so that it is more difficult to reduce food inventory. Thus, the government can hardly continue to use huge funds to adjust grain supply. Moreover, the high grain cost would cause a lot of foreign grain to pour into the domestic market. According to the import and export trade data released by General Administration of Customs, in 2021, China's annual grain imports reached 166.94 million tons, China became the largest grain importing country, and China's imports of crops such as soybeans, wheat and corns ranked among the world's highest. The increase in the grain cost leads to a sharp increase in the grain imports, thereby forming a vicious circle of high grain yield, high imports and high inventory [67].

### 4.3. Grain Planting Structure Does Not Match Consumption Structure

In the new development stage, the main contradiction faced by Chinese society has turned into the contradiction between unbalanced and inadequate development and the people's ever-growing needs for a better life. As income rises, the grain consumption structure of residents is continuously upgraded, and people are no longer satisfied by "enough to eat", but "being well fed" and "healthy eating", and people's demand for safe, green, environmentally friendly and nutritious food keeps growing. However, at present, the green high-quality agricultural products in China are in short supply, and the problem faced by food security has turned into the structural contradiction under resource constraints from insufficient quantity [68]. Seen from the angle of supply, China's food supply is at risk of becoming greater than demand on the whole and less than demand in some areas. In terms of the grain ration, China's grain ration is absolutely safe at present. According to the data released in the China Agricultural Outlook Report (2022–2031), in 2021, China's grain consumption was 31,617 million tons, the actual grain yield was 66.234 million tons, and the grain ration guarantee degree was 197%. In terms of the structure, the main grain crops such as rice, wheat and corns are oversupplied in some stages, and the inventory is high. High-quality wheat products (e.g., plain flour and strong flour), minor grain crops and high-quality rice are undersupplied and need to be imported from foreign countries.

In terms of feed grain, with the development of urbanization, the consumption structure of Chinese people has turned into the consumption of animal food such as meat, eggs, milk and aquatic products from the consumption of staple food grain, and the gap in protein feed grain is big. On the whole, there are great differences among various grain varieties in China in terms of self-sufficiency rate. Some varieties are oversupplied, some varieties are undersupplied, and the structural contradiction between grain production and supply and consumer demand creates great risks for the future food security [69]. Furthermore, due to the impact of decentralization of producers, the grain production structure in China is mainly manifested as follows: (i). structural assimilation in variety, in the centralized producing area of wheat, rough rice, corns and soybeans, the grain yield of the same variety is high; and (ii). structural assimilation in quality, the producers lack of the ability to produce grain food according to requirements. To some extent, assimilation in grain production means homogenization and is manifested as low level in China's food structure security.

Moreover, the prominent grain loss and wastage problem in China, plus the factors such as unreasonable consumption structure, further increase the pressure in terms of food security. On one hand, loss and wastage results in the big gap between the demand and

the share of grain; on the other hand, unreasonable dietary structure is also an important factor that threatens China's food security [1]. Restricted by the availability of data, this article does not consider grain loss and wastage and food consumption structure. The researcher plans to seek the evaluation indicators that can reflect the two factors in the subsequent research process and conduct empirical research on them. In addition, this study used the entropy weighting method to derive the weights of each evaluation index, and no comparative analysis was conducted with the results obtained from other weighting calculation methods. In the subsequent study, it is planned to use hierarchical analysis, principal component analysis, and other methods to calculate the weights of each index separately and conduct comparative analysis to arrive at the most scientific and reasonable index weights.

## 5. Conclusions

Food security is the foundation and prerequisite for social stability and economic development. The evaluation of food security status and the analysis of food security situation can provide basis for the scientific selection of the food security strategy in the new development stage and the promotion of sustainable development of agriculture. This article starts from quantity security, structural security, ecological security of resources, economic security and policy security and conducts empirical research on the state of China's food security from 2001 to 2020. The results show that: (1) from 2001 to 2020, China's food security level shows a trend of declining firstly and then rising; (2) China attaches increasing importance to the ecological security and policy security of food resources; (3) the promotion of China's food security level is restricted by such factors as resources, mode of production, circulation, storage and transportation, trade and structure. Accordingly, this article puts forward relevant policy suggestions as follows:

Implement the strategy of storing grain in land, stick to the red line in terms of the quantity and quality of arable land, and safeguard the grain production capacity. Firstly, the Chinese government should strictly observe the red line of 1.8 billion mu of arable land, attach importance to increasing land consolidation and reclamation, implement the replenishment of arable land, avoid the extensive use or abandonment of arable land, and prevent arable land from becoming non-agricultural or non-grain agricultural areas. Meanwhile, it is also crucial to increase investment in agricultural infrastructure and improve field irrigation and ecological protection projects, and build about 1 billion mu of high-standard farmland with the focus on major grain-producing areas. Thirdly, it also needs to promote green agricultural production activities, such as reduced application of chemical fertilizers and pesticides, soil testing and formula fertilization techniques, and conservation tillage techniques to steadily improve the quality of cultivated land. Finally, China should strengthen the construction of systems for monitoring basic farmland area and soil quality, and improve China's monitoring and early warning capacity for the area and quality of cultivated land.

Implement the strategy of storing grain in technology, increase the content of agricultural science and technology, and improve the comprehensive competitiveness of agriculture. Increase inputs in agricultural science and technology and carry out a revolution of production technologies to realize "storing grain in technology", to promote the advances in and the popularization of agricultural technology. Strengthen the transformation and application of scientific and technological achievements of seed-breeding techniques, production techniques, mechanical techniques and information techniques. Coordinate the relationship between the grain production and the ecological environment, conduct R and D activities to popularize the technique of returning straw to field and the biological pesticide technique, reduce the agricultural non-point source pollution, and enhance the capacity for the sustainable development of agriculture. Improve the level of information technology in the grain industry and rely on big data and advanced technologies in the internet of things to promote the transformation and upgrading of the whole industry chain covering grain production, processing, circulation and storage. Strengthen the construction

of the innovation system for grain science and technology, integrate scientific and technological resources, adjust the regional layout, clarify the division of labor and cooperation, optimize the innovation environment, and enhance the innovation capacity of grain science and technology.

Coordinate and utilize two resources (i.e., international and domestic resources) and two markets (i.e., international and domestic markets), and strengthen deep cooperation with foreign countries On the basis of adhering to absolute food security and basic self-sufficiency in cereals, China should expand trade channels through diversified import sources to maintain the stability of imports for soybeans and coarse grains that do not have comparative advantages in production. Use international markets and resources to stimulate domestic production potential, optimize the layout of productive forces, concentrate advantageous resource conditions, and strengthen the construction of advantageous industrial zones, highlighting major grain varieties such as rice and wheat. Actively participate in the international governance system of grain trade and the grain acquisition and distribution chain in developing countries to improve the discourse and China's international position in the global grain market.

**Author Contributions:** Conceptualization, X.Z.; methodology, T.W.; software, T.W.; validation, J.B.; formal analysis, Y.W.; writing—original draft preparation, X.Z. and Y.W.; writing—review and editing, J.B.; supervision, Y.W.; funding acquisition, S.X. All authors have read and agreed to the published version of the manuscript.

**Funding:** This research was funded by The Science and Technology Innovation Project of the Chinese Academy of Agricultural Sciences (CAAS-ASTIP-2022-AII).

**Institutional Review Board Statement:** Not applicable.

**Data Availability Statement:** The data presented in this study are available within the article.

**Conflicts of Interest:** The authors declare no conflict of interest.

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
