# Peer review of "A Research on the Evaluation of China’s Food Security under the Perspective of Sustainable Development—Based on an Entropy Weight TOPSIS Model"

_agriculture, doi:10.3390/agriculture12111926_

Round 1
Reviewer 1 Report
Abstract
The research method and data used, the time period studied, need to be presented.
Introduction
It is suggested that the authors conduct a thorough review of the research literature in order to identify research gaps and the contribution of this research. It is preferable to have a separate section for research literature.
Authors should improve references in accordance with current literature and the journal's style.
Methods and Data
It is recommended that the source of each index and sub-index in Table 1 be provided separately.
Engel's Coefficient of Rural Residents? Is the data of this index available?
It appears that the productivity of scarce inputs is critical. Is labour a scarce resource in China? Water productivity, in my opinion, is critical due to China's water scarcity, and even energy productivity should be considered.
The entropy weighting method is simply a statistical tool that determines weights based on data dispersion. As a result, it cannot produce consistent results. It is highly recommended to calculate the weights using the AHP method and compare the results.
Results
Non-normalized weights should also be reported.
Are the results related to the TOPSIS method reported in the results section?
It is recommended to evaluate the food security situation by ranking the years using the TOPSIS method.
Conclusion
The policy implications need to be more concrete.
Author Response
The authors have revised the article according to the reviewers' comments, and the details of the revisions are attached.

Reviewer 2 Report
Authors can add a comparison of TOPSIS/Entropy weight TOPSIS methods with other weight-assigning methods to the various groups of indicators.
Author Response

(The authors gave the same response as above.)

Reviewer 3 Report
As limited arable land availability, deteriorating soil quality, water scarcity and pollution, climate change, and intensive reliance on fertilizers and pesticides have become widespread issues, they will all begin to limit or reduce agricultural production in China, thereby threatening food security.
The research begins with the new connotation and objectives of food security in the new development stage, takes quantity security, structural security, ecological security of resources, economic security, and policy security as the breakthrough points, constructs a food security evaluation system with 25 indicators, and aims to conduct evaluation and research on China's food security's evolution and current state (from 2001 to 2020).
Introduction
No comments. The introduction presents background information and purpose of study.
There are no remarks regarding the structure.
Materials and Methods
· The methods employed are explained in detail. The variables used in the model are defined and explained. There are no observations
Results/ Discussion
Within figure no. 1 the 2nd degree indicators are not mentioned at all on the x-axis.
Conclusion
There are no observations
Author Response

(The authors gave the same response as above.)

Round 2
Reviewer 1 Report
It is recommended that the following comments be addressed.
1. It is necessary to report the results of the last stage of TOPSIS (Index Si -equation 17).
2. It is recommended to evaluate the food security situation by ranking the years using the TOPSIS method.
Author Response
Dear reviewers,
the authors have revised the article according to the review comments, please see the attachment for details of the revisions.
Thank you very much.
